# Spatial Broadcast Decoder: A Simple Architecture for Disentangled Representations in VAEs

## Abstract

We present a neural rendering architecture that helps variational autoencoders (VAEs) learn disentangled representations. Instead of the deconvolutional network typically used in the decoder of VAEs, we tile (broadcast) the latent vector across space, concatenate fixed X- and Y-"coordinate" channels, and apply a fully convolutional network with $1 \times 1$ stride. This provides an architectural prior for dissociating positional from non-positional features in the latent space, yet without providing any explicit supervision to this effect. We show that this architecture, which we term the *Spatial Broadcast decoder*, improves disentangling, reconstruction accuracy, and generalization to held-out regions in data space. We show the Spatial Broadcast Decoder is complementary to state-of-the-art (SOTA) disentangling techniques and when incorporated improves their performance.

## 1 Introduction

Knowledge transfer and generalization are hallmarks of human intelligence. From grammatical generalization when learning a new language to visual generalization when interpreting a Picasso, humans have an extreme ability to recognize and apply learned patterns in new contexts. Current machine learning algorithms pale in contrast, suffering from overfitting, adversarial attacks, and domain specialization (Lake et al., 2016; Marcus, 2018). We believe that one fruitful approach to improve generalization in machine learning is to learn compositional representations in an unsupervised manner. A compositional representation consists of components that can be recombined, and such recombination underlies generalization. For example, consider a pink elephant. With a representation that composes color and object independently, imagining a pink elephant is trivial. However, a pink elephant may not be within the scope of a representation that mixes color and object. Compositionality comes in a variety of flavors, including feature compositionality (e.g. pink elephant), multi-object compositionality (e.g. elephant next to a penguin), and relational compositionality (e.g. the smallest elephant). In this work we focus on feature compositionality.

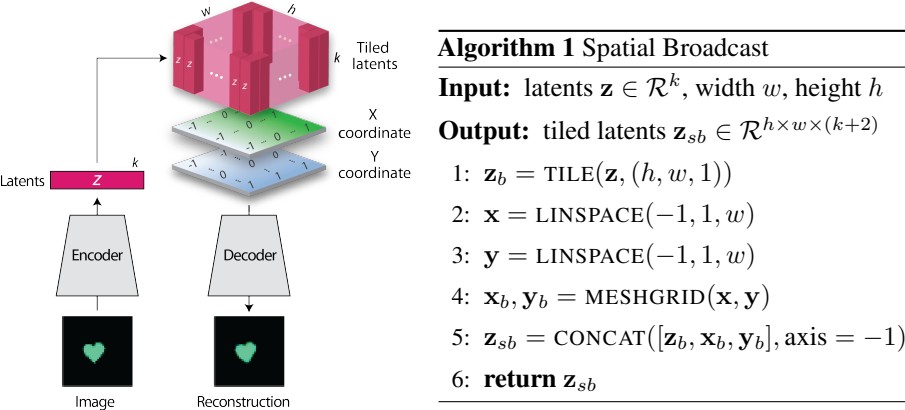

**Algorithm 1** Spatial Broadcast

**Input:** latents $\mathbf{z} \in \mathcal{R}^k$, width $w$, height $h$

**Output:** tiled latents $\mathbf{z}_{sb} \in \mathcal{R}^{h \times w \times (k+2)}$

1: $\mathbf{z}_b = \text{TILE}(\mathbf{z}, (h, w, 1))$

2: $\mathbf{x} = \text{LINSPACE}(-1, 1, w)$

3: $\mathbf{y} = \text{LINSPACE}(-1, 1, w)$

4: $\mathbf{x}_b, \mathbf{y}_b = \text{MESHGRID}(\mathbf{x}, \mathbf{y})$

5: $\mathbf{z}_{sb} = \text{CONCAT}([\mathbf{z}_b, \mathbf{x}_b, \mathbf{y}_b], \text{axis} = -1)$

6: **return** $\mathbf{z}_{sb}$

**Figure 1:** *(left)* Schematic of the Spatial Broadcast VAE. In the decoder, we broadcast (tile) a latent sample to the image width and height, and concatenate two "coordinate" channels. This is then fed to an unstrided convolutional decoder. *(right)* Pseudo-code of the spatial broadcast operation, assuming a `numpy` / Tensorflow-like API.

Representations with feature compositionality are sometimes referred to as "disentangled" representations (Bengio et al., 2013). Learning disentangled representations from images has recently garnered much attention. However, even in the best understood conditions, finding hyperparameters to robustly obtain such representations still proves quite challenging (Locatello et al., 2018). Here we present the *Spatial Broadcast decoder* (Figure 1), a simple modification of the variational autoencoder (VAE) (Kingma and Welling, 2014; Rezende et al., 2014) decoder architecture that:

- Improves reconstruction accuracy and disentangling in a VAE on datasets of simple objects.
- Is complementary to (and improves) state-of-the-art disentangling techniques.
- Shows particularly significant benefits when the objects in the dataset are small, a regime notoriously difficult for standard VAE architectures (Appendix D).
- Improves representational generalization to out-of-distribution test datasets involving both interpolation and extrapolation in latent space (Appendix H).

## 2    SPATIAL BROADCAST DECODER

When applying VAEs to image datasets, standard architecture decoders consist of an MLP followed by an upsampling deconvolutional network. However, with this architecture a VAE learns highly entangled representations in an effort to represent the data as closely as possibly to its Gaussian prior (e.g. Figure 4). A number of new variations of the VAE objective have been developed to encourage disentanglement, though all of them introduce additional hyperparameters (Higgins et al., 2017a; Burgess et al., 2018; Kim and Mnih, 2017; Chen et al., 2018). Furthermore, a recent study found them to be extremely sensitive to these hyperparameters (Locatello et al., 2018).

Meanwhile, upsampling deconvolutional networks like the standard VAE decoder have been found to pose optimization challenges, such as checkerboard artifacts (Odena et al., 2016) and spatial discontinuities (Liu et al., 2018). These effects seem likely to raise problems for latent space representation-learning. Intuitively, asking a deconvolutional network to render an object at a particular position is a tall order: Since the network's filters have no explicit spatial information, the network must learn to propagate spatial asymmetries down from its highest layers and in from the edges of the layers. This is a complicated function to learn, so optimization is difficult. To remedy this, in the Spatial Broadcast decoder (Figure 1) we remove all upsampling deconvolutions from the network, instead tiling the latent vector across space, appending fixed coordinate channels, then applying an unstrided convolutional network. With this architecture, rendering an object at a position becomes a very simple function. Such simplicity of computation gives rise to ease of optimization.

In addition to better disentangling, we find that the Spatial Broadcast VAE can yield better reconstructions (Figure 3), all with shallower networks and fewer parameters than a standard deconvolutional architecture. However, it is worth noting that a standard DeConv decoder may in principle more easily place patterns relative to each other or capture more extended spatial correlations. We did not find this to impact performance, even on datasets with extended sparial correlations and no variation of object position (Appendix F), but it is still a possible limitation of our model.

The idea of appending coordinate channels to convolutional layers has recently been highlighted (and named CoordConv) in the context of improving positional generalization (Liu et al., 2018). However,the CoordConv technique had been used beforehand (Zhao et al., 2015; Liang et al., 2015; Watters et al., 2017; Wojna et al., 2017; Perez et al., 2017; Ulyanov et al., 2017) and its origin is unclear. While CoordConv VAE (Liu et al., 2018) incorporates CoordConv layers into an upsampling deconvolutional network in a VAE (see Appendix E), to our knowledge no prior work has combined CoordConv with spatially tiling a generative model's representation as we do here.

## 3    RESULTS

The Spatial Broadcast decoder was designed with object-feature representations in mind, so to initially showcase its performance we use a dataset of simple objects: colored 2-dimensional sprites (Burgess et al., 2018; Matthey et al., 2017). This dataset has 8 factors of variation: X-position, Y-position, Size, Shape, Angle, and three-dimensional Color.

In Figure 2 we compare a standard DeConv VAE (a VAE with an MLP + deconvolutional network decoder) to a Spatial Broadcast VAE (a VAE with the Spatial Broadcast decoder). We see that

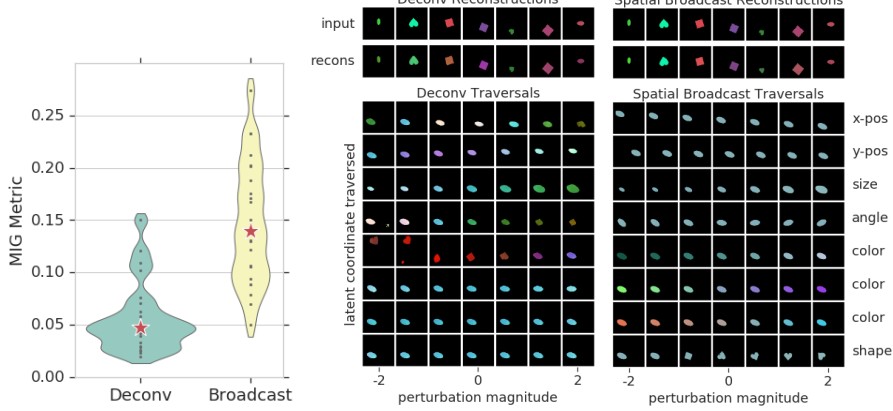

**Figure 2: Comparing DeConv to Spatial Broadcast decoder in a VAE.** *(left)* MIG results for 25 random seeds, showing a Spatial Broadcast VAE achieves better scores than a DeConv VAE. Stars are median MIG values and the seeds used for the traversals to the right. *(middle)* DeConv VAE reconstructions and latent space traversals. Traversals are generated around a seed point in latent space by reconstructing a sweep from -2 to +2 for each coordinate while keeping all other coordinates constant. The traversal shows entanglement. *(right)* Spatial Broadcast VAE reconstructions and traversal. The traversal is disentangled and aligned with generative factors (labels on the right attributed by visual inspection). While all models were trained with 10 latent coordinates, only the 8 lowest-variance ones are shown in the traversals (the other two are non-coding).

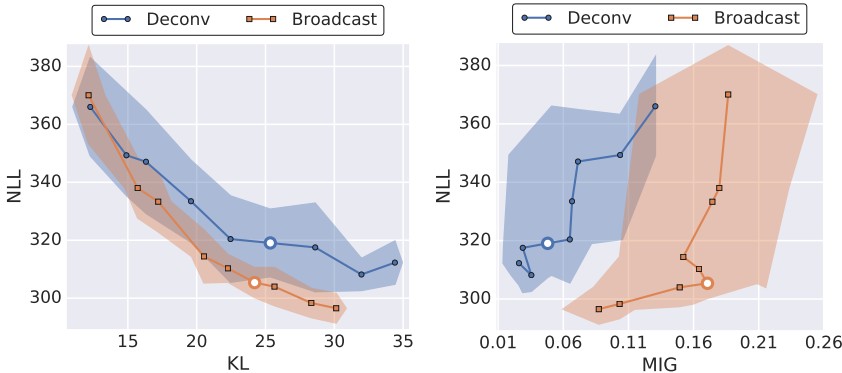

**Figure 3: Rate-distortion curves.** We swept $\beta$ log-linearly from 0.4 to 5.4 and for each value trained 10 replicas each of Deconv $\beta$-VAE (blue) and Spatial Broadcast $\beta$-VAE (orange) on colored sprites. The dots show the mean over these replicas for each $\beta$, and the shaded region shows the hull of one standard deviation. White dots indicate $\beta = 1$. *(left)* Reconstruction (Negative Log-Likelihood, NLL) vs KL. $\beta < 1$ yields low NLL and high KL (bottom-right of figure), whereas $\beta > 1$ yields high NLL and low KL (top-left of figure). See Alemi et al. (2017) for details. Spatial Broadcast $\beta$-VAE shows a better rate-distortion curve than Deconv $\beta$-VAE. *(right)* Reconstruction vs MIG metric. $\beta < 1$ correspond to lower NLL and low MIG regions (bottom-left of figure), and $\beta > 1$ values correspond to high NLL and high MIG scores (towards top-right of figure). Spatial Broadcast $\beta$-VAE is better disentangled (higher MIG scores) than Deconv $\beta$-VAE.

the Spatial Broadcast VAE outperforms the DeConv VAE both in terms of the Mutual Information Gap (MIG) disentangling metric (Chen et al., 2018) and traversal visualizations, even though the hyperparameters for the models were chosen to minimize the model's error, not chosen for any disentangling properties explicitly (details in Appendix G).

The Spatial Broadcast decoder is complementary to existing disentangling VAE techniques, hence improves not only a vanilla VAE but SOTA models as well. For example, Figure 3 shows that the Spatial Broadcast decoder improves disentangling and yields a lower rate-distortion curve in a $\beta$-VAE (Higgins et al., 2017a), hence induces a more efficient representation of the data than a DeConv decoder. See Appendix C for similar results in other SOTA models.

Evaluating the quality of a representation can be challenging and time-consuming. While a number of metrics have been proposed to quantify disentangling, many of them have serious shortcomings and there is as yet no consensus in the literature which to use (Locatello et al., 2018). Visualizing reconstructions of latent space traversals (Figures 2, 10, 7) can provide a subjective glimpse of the quality of the representation, but only reflects the representation at one point in latent space and can be difficult to interpret by eye. Consequently, we propose an additional latent space analysis method,

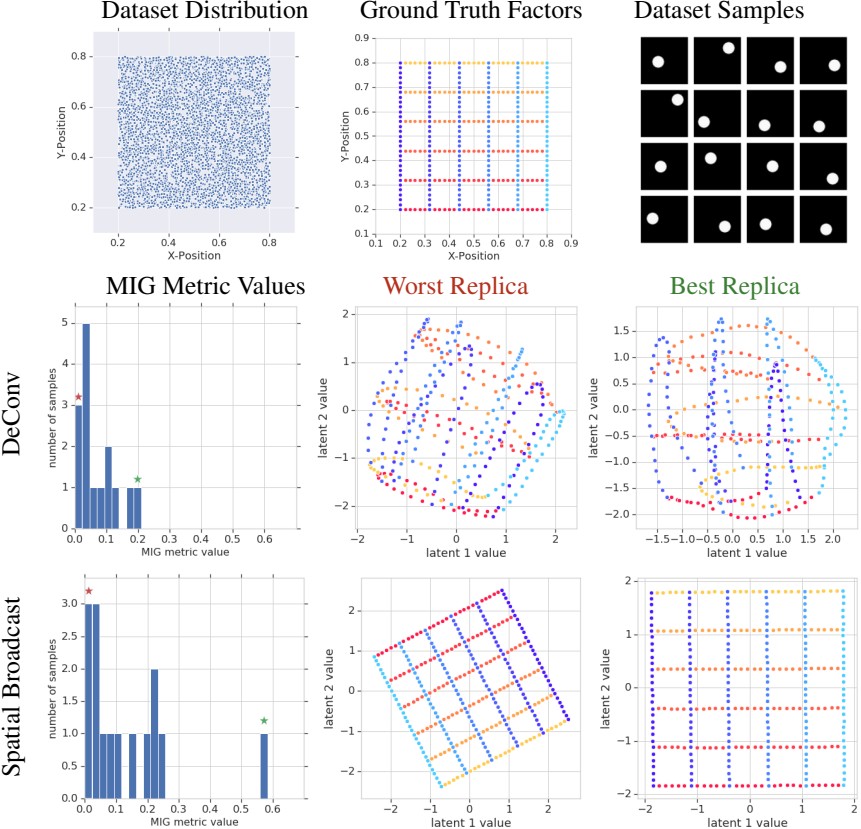

**Figure 4: Latent space geometry analysis of x-y dataset.** Comparison of DeConv and Spatial Broadcast VAEs on a dataset of circles varying in position. On the top row we show: *(left)* the data generative factor distribution uniform in X- and Y-position, *(middle)* a grid of points in generative factor space spanning the data distribution, and *(right)* 16 sample images from the dataset. The next two rows show a analysis of the DeConv VAE and Spatial Broadcast VAE on this dataset. In each we see a histogram of MIG metric values over 15 independent replicas and a latent space geometry visualization for the replica with the worst MIG and the replica with the best MIG (colored stars in the histograms). These geometry plots show the embedding of the ground truth factor grid in the latent subspace spanned by the two lowest-variance (most informative) latent components. Note that the MIG does not capture this contrast because it is very sensitive to rotation in the latent space (see Appendex H for more discussion about the MIG metric).

which we found useful in our research: We plot in latent space the locations corresponding to a grid of points in generative factor space, thereby viewing the embedding of generative factor space in the model's latent space. While this is only shows the latent embedding of a 2-dimensional subspace of generative factor space, it can be very revealing of the latent space geometry.

We showcase this analysis method in Figure 4 on a dataset of circles varying in X- and Y-position. This shows that a Spatial Broadcast VAE's learned latent representation is a near-perfect Euclidean transformation of the data generative factors, in sharp contrast to a DeConv VAE's representation. This disentangling with the Spatial Broadcast decoder is robust even when the generative factors in the dataset are not independent (Appendix H).

# 4 CONCLUSION

Here we present the Spatial Broadcast decoder for Variational Autoencoders. We demonstrate that it improves learned latent representations, most dramatically for datasets with objects varying in position. It also improves generalization in latent space and can be incorporated into SOTA models to boost their performance in terms of both disentangling and reconstruction accuracy. We believe that learning compositional representations is an important ingredient for flexibility and generalization in many contexts, from supervised learning to reinforcement learning, and the Spatial Broadcast decoder is one step towards robust compositional visual representation learning.

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

# Supplementary material

## A  EXPERIMENT DETAILS

For all VAE models we used a Bernoulli decoder distribution, parametrized by its logits. It is with respect to this distribution that the reconstruction error (negative log likelihood) was computed. This could accomodate our datasets, since they were normalized to have pixel values in $[0, 1]$. We also explored using a Gaussian distribution with fixed variance (for which the NLL is equivalent to scaled MSE), and found that this produces qualitatively similar results and in fact improves stability. Hence while a Bernoulli distribution usually works, we suggest the reader wishing to experiment with these models starts with a Gaussian decoder distribution with mean parameterized by the decoder network output and variance constant at $0.3$.

In all networks we used ReLU activations, weights initialized by a truncated normal (Ioffe and Szegedy, 2015), and biases initialized to zero. We use no other neural network tricks (no BatchNorm or dropout), and all models were trained with the Adam optimizer (Kingma and Ba, 2015). See below for learning rate details.

### A.1  VAE HYPERPARAMETERS

For all VAE models except $\beta$-VAE (shown only in Figure 3), we use a standard VAE loss, namely with a KL term coefficient $\beta = 1$. For FactorVAE we also use $\beta = 1$, as in Kim and Mnih (2017).

For the VAE, $\beta$-VAE, CoordConv VAE and ablation study we used the network parameters in Table 1. We note that, while the Spatial Broadcast decoder uses fewer parameters than the DeConv decoder, it does require about $50\%$ more memory to store the weights. However, for the 3D Object-in-Room dataset we included three additional deconv layers in the Spatial Droadcast decoder (without these additional layers the decoder was not powerful enough to give good reconstructions on that dataset).

All of these models were trained using a learning rate of $3 \cdot 10^{-4}$ on with batch size 16. All convolutional and deconvolutional layers have "same" padding, i.e. have zero-padded input so that the output shape is `input_shape × stride` in the case of convolution and `input_shape/stride` in the case of deconvolution.

| ENCODER |
| --- |
| `FC(2 × 10) Output LogNormal` |
| `FC(256)` |
| `Conv(k=4, s=2, c=64)` |
| `Conv(k=4, s=2, c=64)` |
| `Conv(k=4, s=2, c=64)` |
| `Conv(k=4, s=2, c=64)` |
| `Input Image [64 × 64 × C]` |

| DECONV DECODER |
| --- |
| `Output Logits` |
| `DeConv(k=4, s=2, c=C)` |
| `DeConv(k=4, s=2, c=64)` |
| `DeConv(k=4, s=2, c=64)` |
| `DeConv(k=4, s=2, c=64)` |
| `DeConv(k=4, s=2, c=64)` |
| `reshape(2 × 2 × 64)` |
| `FC(256)` |
| `Input Vector [10]` |

| BROADCAST DECODER |
| --- |
| `Output Logits` |
| `Conv(k=4, s=1, c=C)` |
| `Conv(k=4, s=1, c=64)` |
| `Conv(k=4, s=1, c=64)` |
| `append coord channels` |
| `tile(64 × 64 × 10)` |
| `Input Vector [10]` |

**Table 1:** Network architectures for Vanilla VAE, $\beta$-VAE, CoordConv VAE and ablation study. The numbers of layers were selected to minimize the ELBO loss of a VAE on the colored sprites data (see Appendix G). Note that for 3D Object-in-Room, we include three additional convolutional layers in the spatial broadcast decoder. Here $C$ refers to the number of channels of the input image, either 1 or 3 depending on whether the dataset has color.

### A.2  FACTORVAE

For the FactorVAE model, we used the hyperparameters described in the FactorVAE paper (Kim and Mnih, 2017). Those network parameters are reiterated in Table 2. Note that the Spatial Broadcast parameters are the same as for the other models in Table 1. For the optimization hyperparameters we used $\gamma = 35$, a learning rate of $10^{-4}$ for the VAE updates, a learning rate of $2 \cdot 10^{-5}$ for the discriminator updates, and batch size 32. These parameters generally gave the best results.

However, when training the FactorVAE model on colored sprites we encountered instability during training. We subsequently did a number of hyperparameter sweeps attempting to improve stability, but to no avail. Ultimately, we used the hyperparameters in Table 2, though even with limited training

steps (see Appendix Section A.4) about $15\%$ of seeds diverged before training completed for both Spatial Broadcast and Deconv decoder.

| ENCODER |
|---|
| FC$(2 \times 10)$  Output LogNormal |
| FC$(256)$ |
| Conv(k=4, s=2, c=64) |
| Conv(k=4, s=2, c=64) |
| Conv(k=4, s=2, c=32) |
| Conv(k=4, s=2, c=32) |
| Input Image $[64 \times 64 \times C]$ |

| DECONV DECODER |
|---|
| Output Logits |
| DeConv(k=4, s=2, c=$C$) |
| DeConv(k=4, s=2, c=32) |
| DeConv(k=4, s=2, c=32) |
| DeConv(k=4, s=2, c=64) |
| DeConv(k=4, s=2, c=64) |
| reshape$(2 \times 2 \times 64)$ |
| FC$(256)$ |
| Input Vector $[10]$ |

| BROADCAST DECODER |
|---|
| Output Logits |
| Conv(k=4, s=1, c=$C$) |
| Conv(k=4, s=1, c=64) |
| Conv(k=4, s=1, c=64) |
| append coord channels |
| tile$(64 \times 64 \times 10$ |
| Input Vector $[10]$ |

**Table 2:** Network architectures for FactorVAE. The encoder and DeConv decoder architectures are as described in the FactorVAE paper (Kim and Mnih, 2017), and the Spatial Broadcast decoder architecture is the same as for the other models (Table 1.)

## A.3 DATASETS

All datasets were rendered in images of size $64 \times 64$ and normalized to $[0, 1]$.

**Colored Sprites:**

For this dataset, we use the binary dsprites dataset open-sourced in (Matthey et al., 2017), but multiplied by colors sampled in HSV space uniformly within the region $H \in [0.0, 1.0]$, $S \in [0.3, 0.7]$, $V \in [0.3, 0.7]$. Sans color, there are 737,280 images in this dataset. However, we sample the colors online from a continuous distribution, effectively making the dataset size infinite.

**Chairs:**

This dataset is open-sourced in (Aubry et al., 2014). This dataset, unlike all others we use, has only a single channel in its images. It contains 86,366 images.

**3D Object-in-Room:**

This dataset was used extensively in the FactorVAE paper (Kim and Mnih, 2017). It consists of an object in a room and has 6 factors of variation: Camera angle, object size, object shape, object color, wall color, and floor color. The colors are sampled uniformly from a continuous set of hues in the range $[0.0, 0.9]$. This dataset contains 480,000 images, procedurally generated as all combinations of 10 floor hues, 10 wall hues, 10 object hues, 8 object sizes, 4 object shapes, and 15 camera angles.

**Circles Datasets:**

To more thoroughly explore datasets with a variety of distributions, factors of variation, and held-out test sets we wrote our own procedural image generator for circular objects in PyGame (rendered with an anti-aliasing factor of 5). We used this to generate the data for results in Figure 4 and Appendix H. In these datasets we control subsets of the following factors of variation: X-position, Y-position, Size, Color. We generated five datasets in this way, which we call X-Y, X-H, R-G, X-Y-H Small, and X-Y-H Tiny, and can be seen in Figures (Fig 11), (Fig 14), (Fig 17), (Fig 7), and (Fig 8) respectively.

Table 3 shows the values of these factors for each dataset. Note that for some datasets we define the color distribution in RGB space, and for others we define it in HSV space.

To create the datasets with dependent factors, we hold out one quarter of the dataset (the intersection of half of the ranges of each of the two factors), either centered within the data distribution or in the corner.

For each dataset we generate 500,000 randomly sampled training images.

| | X | Y | Size | (H, S, V) | (R, G, B) |
|---|---|---|---|---|---|
| X-Y | [0.2, 0.8] | [0.2, 0.8] | 0.2 | N/A | (1.0, 1.0, 1.0) |
| X-H | [0.2, 0.8] | 0.5 | 0.3 | ([0.2, 0.8], 1.0, 1.0) | N/A |
| R-G | 0.5 | 0.5 | 0.5 | N/A | ([0.4, 0.8], [0.4, 0.8], 1.0) |
| X-Y-H Small | [0.2, 0.8] | [0.2, 0.8] | 0.1 | ([0.2, 0.8], 1.0, 1.0) | N/A |
| X-Y-H Tiny | [0.2, 0.8] | [0.2, 0.8] | 0.075 | ([0.2, 0.8], 1.0, 1.0) | N/A |

**Table 3:** Circles datasets details. Each row represents a different dataset, and each column shows a generative factor's range for the datasets.

### A.4 TRAINING STEPS

The number of training steps for each model on each dataset can be found in Table 4. In general, for each dataset we used enough training steps so that all models converged. Note that while the training iterations is different for FactorVAE than for the other models on colored sprites (due to instability of FactorVAE), this has no bearing on our results because we do not compare across models. We only compare across decoder architectures, and we always used the same training steps for both DeConv and Spatial Broadcast decoders within each model.

| | VAE | FactorVAE |
|---|---|---|
| COLORED SPRITES | $1.5 \cdot 10^6$ | $1 \cdot 10^6$ |
| CHAIRS | $7 \cdot 10^5$ | N/A |
| 3D OBJECTS | $2.5 \cdot 10^6$ | N/A |
| CIRCLES | $5 \cdot 10^5$ | $5 \cdot 10^5$ |

**Table 4:** Number of training steps for each model on each dataset. Here the VAE column includes $\beta$-VAE, Deconv VAE, Spatial Broadcast VAE, CoordConv VAE, and the ablation study.

## B  ABLATION STUDY

One aspect of the Spatial Broadcast decoder is the concatenation of constant coordinate channels to its tiled input latent vector. While our justification of its performance emphasizes the simplicity of computation it affords, it may seem possible that the coordinate channels are only used to provide positional information and the simplicity of this positional information (linear meshgrid) is irrelevant. Here we perform an ablation study to demonstrate that this is not the case; the organization of the coordinate channels is important. For this experiment, we randomly permute the coordinate channels through space. Specifically, we take the $[h \times w \times 2]$-shape coordinate channels and randomly permute the $h \cdot w$ entries. We keep each $(i, j)$ pair together to ensure that after the shuffling each location does still have a unique pair of coordinates in the coordinate channels. Importantly, we only shuffle the coordinate channels once, then keep them constant throughout training.

Figure 5 shows reconstructions and traversals for two replicas (with different shuffled coordinate channels). Both disentangling and reconstruction accuracy are significantly reduced.

## C  FACTORVAE WITH SPATIAL BROADCAST DECODER

As shown in Figure 3, the Spatial Broadcast decoder improves disentangling and the rate-distortion trade-off for $\beta$-VAE (Higgins et al., 2017a), indicating it improves state-of-the-art models. To further support this indication, we introduce the Spatial Broadcast decoder into the recently developed FactorVAE (Kim and Mnih, 2017).

Indeed, Figure 6 shows the Spatial Broadcast decoder improves disentangling in FactorVAE. See Appendix H for further results to this effect.

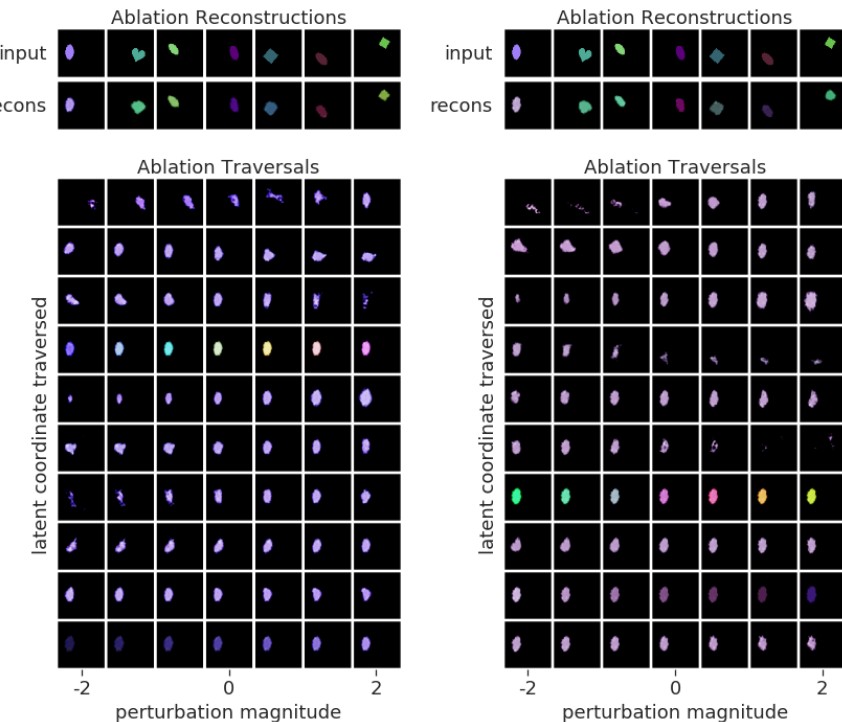

**Figure 5: Ablation Study** Traversals from a Spatial Broadcast VAE with a random (though constant throughout training) shuffling of the coordinate channels. Different model instances (with different shufflings) are show on left and the right. We see that the this shuffling negatively impacts both reconstructions and traversals, hence the linear organization of the coordinate channels is important for the model's performance. Quantitatively, this shuffling causes the MIG to drop from $0.147$ to $0.104 \pm 0.025$ (though still better than a DeConv VAE's $0.052$) and the ELBO loss to drop from $329$ to $362.8 \pm 2.90$ (worse than the DeConv VAE's $347$). The shuffled-coordinate model has KL loss $26.56 \pm 0.47$, number of relevant latents $8.99 \pm 0.37$ and negative log likelihood $336.3 \pm 2.98$.

## D  DISENTANGLING SMALL OBJECTS

In exploring datasets with objects varying in position, we often find a (standard) DeConv VAE learns a representation that is discontinuous with respect to object position. This effect is amplified as the size of the object decreases. This makes sense, because the pressure for a VAE to represent position continuously comes from the fact that an object and a position-perturbed version of itself overlap in pixel space (hence it is economical for the VAE to map noise in its latent samples to local translations of an object). However, as an object's size decreases, this pixel overlap decreases, hence the pressure for a VAE to represent position continuously weakens.

In this small-object regime the Spatial Broadcast decoder's architectural bias towards representing positional variation continuously proves extremely useful. We see this in Figure 7 and Figure 8.

We were surprised to see disentangling of such tiny objects in Figure 8 and have not explored the lower object size limit for disentangling with the Spatial Broadcast decoder.

## E  COORDCONV VAE

CoordConv VAE (Liu et al., 2018) has been proposed as a decoder architecture to improve the continuity of VAE representations. CoordConv VAE appends coordinate channels to every feature layer of the standard deconvolutional decoder, yet does not spatially tile the latent vector, hence retains upsampling deconvolutions.

Figure 9 shows analysis of this model on the colored sprites dataset. While the latent space does appear to be continuous with respect to object position, it is quite entangled (far more so than a Spatial

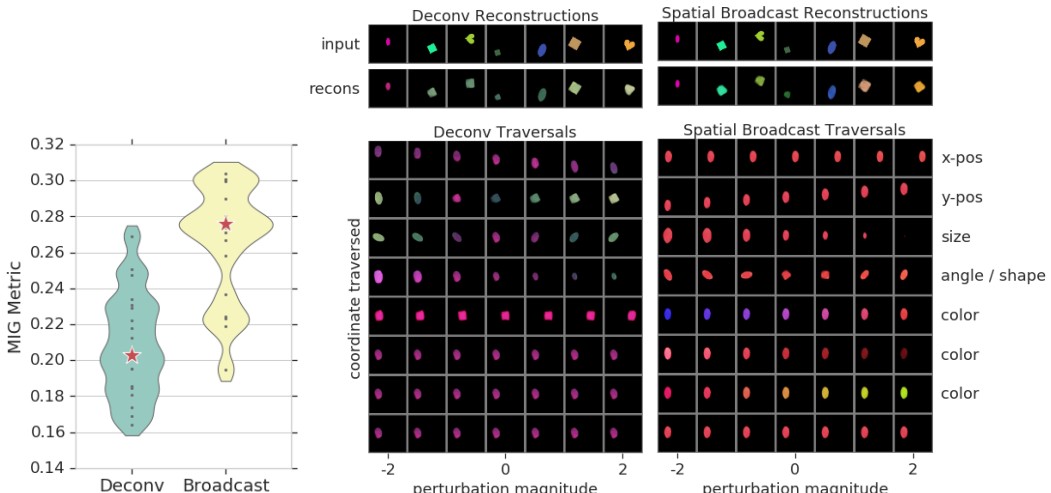

**Figure 6: Comparing Deconv to Spatial Broadcast decoder in a FactorVAE.** *(left)* MIG results, showing a Spatial Broadcast FactorVAE acheives higher (better) scores than a DeConv FactorVAE. Stars are median MIG values and the seeds used for the traversals on the right. *(middle)* DeConv FactorVAE reconstructions and entangled latent space traversals. *(right)* Spatial Broadcast FactorVAE reconstructions and traversal. The traversal is well-disentangled. As in Figure 2, only the most relevant 8 of each model's 10 latent coordinates are shown in the traversals.

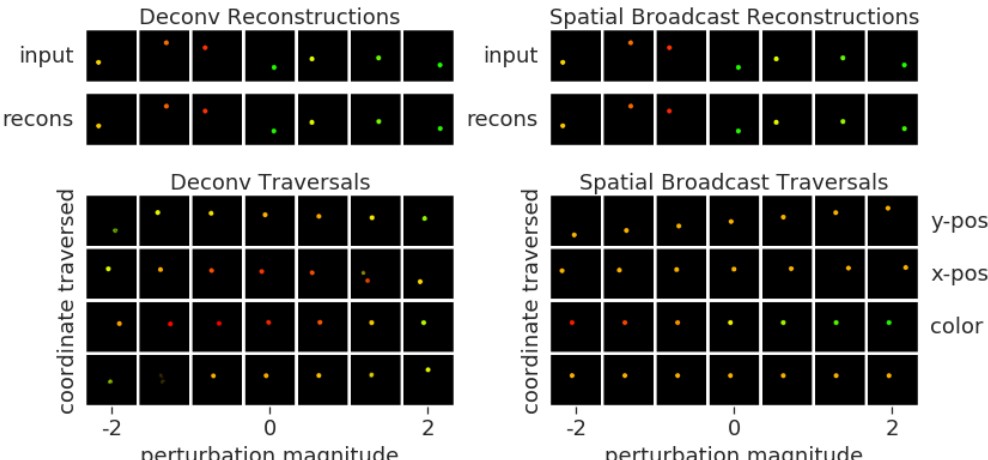

**Figure 7: Disentangling small objects.** A DeConv VAE *(left)* learns a highly entangled and discontinuous representation of this dataset of small hue-varying circles, while a Spatial Broadcast VAE *(right)* disentangles the dataset well. Traversals of only the most relevant 4 of the models' 10 latent coordinates are shown (the other latent coordinates are non-coding). The circles are 0.1 times the frame-width.

Broadcast VAE). This is not very surprising, since CoordConv VAE uses upscale deconvolutions to go all they way from spatial shape $1 \times 1$ to spatial shape $64 \times 64$, while in Table 10 we see that introducing upscaling hurts disentangling in a Spatial Broadcast VAE.

## F    DATASETS WITHOUT POSITIONAL VARIATION

The sprites and circles datasets discussed in Section 3 seem particularly well-suited for the Spatial Broadcast decoder because X- and Y-position are factors of variation. However, we also evaluate the architecture on datasets that have no positional factors of variation: Chairs and 3D Object-in-Room datasets (Aubry et al., 2014; Kim and Mnih, 2017). In the latter, the factors of variation are highly non-local, affecting multiple regions spanning nearly the entire image. We find that on both datasets a Spatial Broadcast VAE learns representations that look as well disentangled as SOTA methods on

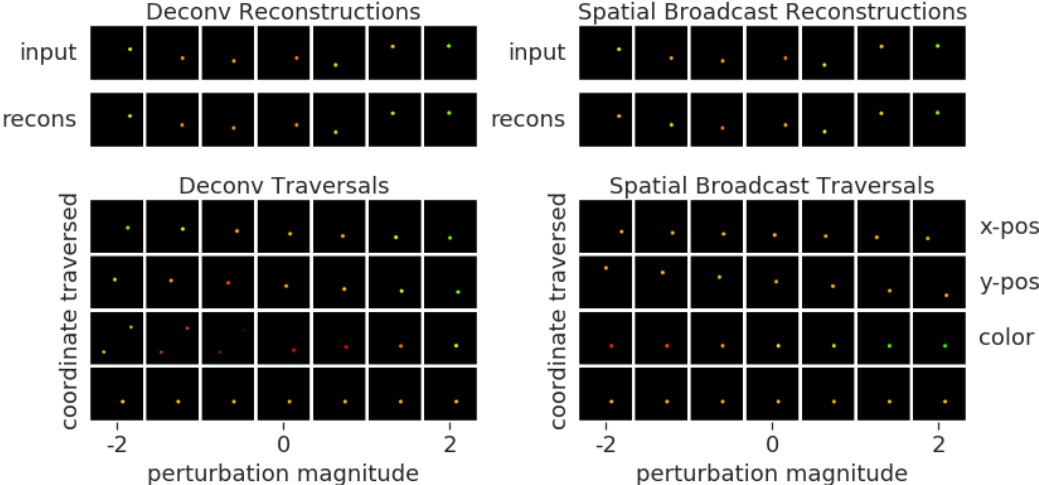

**Figure 8: Disentangling tiny objects.** A DeConv VAE *(left)* learns a highly entangled and discontinuous representation of this dataset of tiny hue-varying circles. In contrast, a Spatial Broadcast VAE *(right)* disentangles this dataset. Traversals of only the most relevant 4 of the models' 10 latent coordinates are shown (the other latent coordinates are non-coding). The circles are 0.075 times the frame-width.

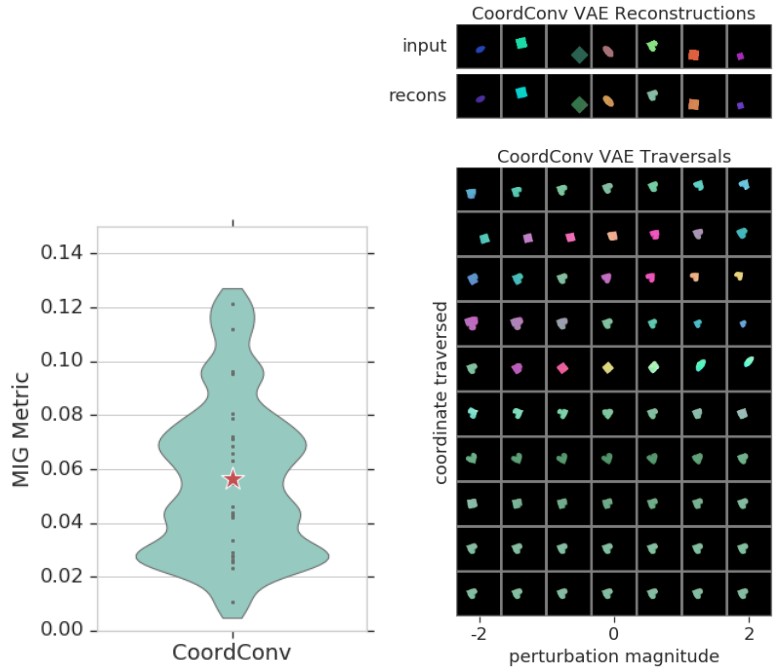

**Figure 9: CoordConv VAE on colored sprites.** *(left)* MIG results. Comparing to Figure 2, we see that this achieves far lower scores than a Spatial Broadcast VAE, though about the same as (or maybe slightly better than) a DeConv VAE. *(right)* Latent space traversals are entangled. Note that in contrast to traversal plots in the main text, we show the effect of traversing all 10 latent components (sorted by smallest to largest mean variance), including the non-coding ones (in the bottom rows).

these datasets and without any modification of the standard VAE objective (Kim and Mnih, 2017; Higgins et al., 2017a). See Figures 10 for results.

We attribute the good performance on these datasets to the Spatial Broadcast Decoder's use of a shallower network and no upsampling deconvolutions (which have been observed to cause optimization difficulties in a variety of settings (Liu et al., 2018; Odena et al., 2016)).

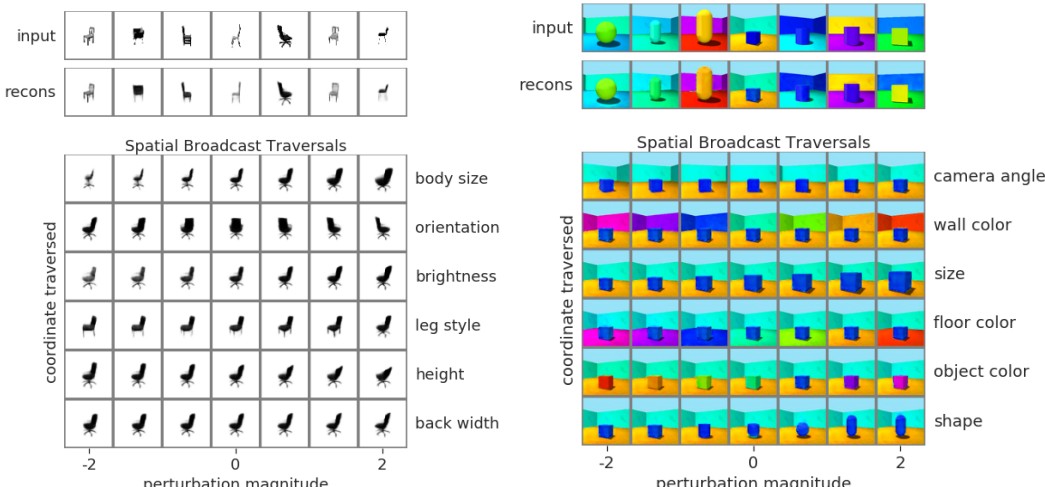

**Figure 10: Traversals for datasets with no positional variation.** A Spatial Broadcast VAE shows good reconstructions and disentangling on the Chairs dataset (Aubry et al., 2014) and the 3D Object-in-Room dataset (Kim and Mnih, 2017).

## G  ARCHITECTURE HYPERPARAMETERS

In order to remain objective when selecting model hyperparameters for the Spatial Broadcast and Deconv decoders, we chose hyperparameters based on minimizing the ELBO loss, not considering any information about disentangling. After finding reasonable encoder hyperparameters, we performed large-scale (25 replicas each) sweeps over a few decoder hyperparameters for both the DeConv and Spatial Broadcast decoder on the colored sprites dataset. These sweeps are revealing of hyperparameter sensitivity, so we report the following quantities for them:

- ELBO. This is the evidence lower bound (total VAE loss). It is the sum of the negative log likelihood (NLL) and KL-divergence.

- NLL. This is the negative log likelihood of an image with respect to the model's reconstructed distribution of that image. It is a measure of reconstruction accuracy.

- KL. This is the KL divergence of the VAE's latent distribution with its Gaussian prior. It measures how much information is being encoded in the latent space.

- Latents Used. This is the mean number of latent coordinates with standard deviation less than $0.5$. Typically, a VAE will have some unused latent coordinates (with standard deviation near $1$) and some used latent coordinates. The threshold $0.5$ is arbitrary, but this quantity does provide a rough idea of how many factors of variation the model may be representing.

- MIG. The MIG metric.

- Factor VAE. This is the metric described in the FactorVAE paper (Kim and Mnih, 2017). We found this metric to be less consistent than the MIG metric (and equally flawed with respect to rotated coordinates), but it qualitatively agrees with the MIG metric most of the time.

### G.1  CONVNET DEPTH

Table 5 shows results of sweeping over ConvNet depth in the Spatial Broadcast decoder. This reveals a consistent trend: As the ConvNet deepens, the model moves towards lower rate/higher distortion. Consequently, latent space information and reconstruction accuracy drop. Traversals with deeper nets show the model dropping factors of variation (the dataset has 8 factors of variation).

Table 6 shows a noisier but similar trend when increasing DeConvNet depth in the DeConv decoder.

| CONVNET | ELBO | NLL | KL | LATENTS USED | MIG | FACTOR VAE |
|---|---|---|---|---|---|---|
| 2-LAYER | 339 ($\pm$2.3) | 312 ($\pm$2.7) | 27.5 ($\pm$0.54) | 8.33 ($\pm$0.37) | 0.076 ($\pm$0.038) | 0.187 ($\pm$0.027) |
| 3-LAYER | **329 ($\pm$4.1)** | 305 ($\pm$4.6) | 24.4 ($\pm$0.54) | 7.22 ($\pm$0.34) | 0.147 ($\pm$0.057) | 0.208 ($\pm$0.084) |
| 4-LAYER | 341 ($\pm$6.8) | 318 ($\pm$7.7) | 22.6 ($\pm$0.97) | 5.93 ($\pm$0.51) | 0.157 ($\pm$0.045) | 0.226 ($\pm$0.046) |
| 5-LAYER | 340 ($\pm$8.8) | 317 ($\pm$9.7) | 22.7 ($\pm$0.99) | 5.70 ($\pm$0.37) | 0.173 ($\pm$0.059) | 0.218 ($\pm$0.030) |

Table 5: **Effect of ConvNet depth on Spatial Broadcast VAE performance.** These results use the colored sprites dataset. Traversals with deeper nets show the model dropping factors of variation (usually color first, then angle, shape, size, and position in that order). The increasing metric values with deeper ConvNets belies the fact that the model is encoding fewer of the 8 factors of variation in the dataset.

| CONVNET | ELBO | NLL | KL | LATENTS USED | MIG | FACTOR VAE |
|---|---|---|---|---|---|---|
| 3-LAYER | 372 ($\pm$8.6) | 346 ($\pm$8.9) | 26.8 ($\pm$0.40) | 9.20 ($\pm$0.04) | 0.031 ($\pm$0.018) | 0.144 ($\pm$0.031) |
| 4-LAYER | 349 ($\pm$9.4) | 322 ($\pm$10.0) | 27.1 ($\pm$0.88) | 8.90 ($\pm$0.24) | 0.025 ($\pm$0.015) | 0.139 ($\pm$0.009) |
| 5-LAYER | **340 ($\pm$9.8)** | 314 ($\pm$10.4) | 26.0 ($\pm$1.00) | 7.95 ($\pm$0.64) | 0.056 ($\pm$0.32) | 0.184 ($\pm$0.053) |
| 6-LAYER | 349 ($\pm$15.0) | 326 ($\pm$16.1) | 23.3 ($\pm$1.42) | 6.36 ($\pm$0.81) | 0.056 ($\pm$0.019) | 0.199 ($\pm$0.029) |

Table 6: **Effect of ConvNet depth on DeConv VAE performance.** These results use the colored sprites dataset. Similarly to Table 5, deeper convnets cause the model to represent fewer factors of variation.

## G.2 MLP DEPTH

The Spatial Broadcast decoder as presented in this work is fully convolutional. It contains no MLP. However, motivated by the need for more depth on the 3D Object-in-Room dataset, we did explore applying an MLP to the input vector prior to the broadcast operation. We found that including this MLP had a qualitatively similar effect as increasing the number of convolutional layers on the colored sprited dataset, decreasing latent capacity and giving poorer reconstructions. These results are shown in Table 7.

However, on the 3D Object-in-Room dataset adding the MLP did improve the model when using ConvNet depth 3 (the same as for colored sprites). Results of a sweep over depth of a pre-broadcast MLP are shown in Table 9. As mentioned in Section F, we were able to achieve the same effect by instead increasing the ConvNet depth to 6, but for those interested in computational efficiency using a pre-broadcast MLP may be a better choice for datasets of this sort.

In the DeConv decoder, increasing the MLP layers again has a broadly similar effect as increasing the ConvNet layers, as shown in Table 8.

| MLP | ELBO | NLL | KL | LATENTS USED | MIG | FACTOR VAE |
|---|---|---|---|---|---|---|
| 0-LAYER | **329 ($\pm$4)** | 305 ($\pm$5) | 24.5 ($\pm$0.54) | 7.21 ($\pm$0.34) | 0.147 ($\pm$0.057) | 0.208 ($\pm$0.084) |
| 1-LAYER | 330 ($\pm$6) | 307 ($\pm$6) | 23.9 ($\pm$0.72) | 6.68 ($\pm$0.46) | 0.164 ($\pm$0.043) | 0.200 ($\pm$0.034) |
| 2-LAYER | 349 ($\pm$15) | 327 ($\pm$17) | 21.5 ($\pm$2.13) | 6.03 ($\pm$0.66) | 0.210 ($\pm$0.048) | 0.232 ($\pm$0.045) |
| 3-LAYER | 392 ($\pm$23) | 399 ($\pm$114) | 15.7 ($\pm$2.93) | 4.17 ($\pm$1.23) | 0.160 ($\pm$0.034) | 0.275 ($\pm$0.064) |

Table 7: **Effect of a pre-broadcast MLP on the Spatial Broadcast VAE's performance.** These results use the colored sprites dataset, on which an MLP seems to hurt performance (though as noted in the text this is not always the case on the 3D Object-in-Room dataset).

| MLP | ELBO | NLL | KL | LATENTS USED | MIG | FACTOR VAE |
|---|---|---|---|---|---|---|
| 1-LAYER | **347 ($\pm$13)** | 321 ($\pm$14) | 25.8 ($\pm$1.3) | 7.97 ($\pm$0.72) | 0.052 ($\pm$0.020) | 0.174 ($\pm$0.043) |
| 2-LAYER | 352 ($\pm$17) | 328 ($\pm$18) | 23.7 ($\pm$1.7) | 6.68 ($\pm$0.78) | 0.051 ($\pm$0.024) | 0.196 ($\pm$0.024) |
| 3-LAYER | 365 ($\pm$19) | 345 ($\pm$21) | 19.7 ($\pm$2.4) | 5.24 ($\pm$0.73) | 0.144 ($\pm$0.062) | 0.243 ($\pm$0.043) |

Table 8: **Effect of MLP depth on the DeConv VAE's performance.** Increasing MLP depth seems to have a broadly similar effect as increasing DeConvNet depth, causing the model to represent fewer factors of variation in the data.

| MLP | ELBO | NLL | KL | LATENTS USED | MIG | FACTOR VAE |
|---|---|---|---|---|---|---|
| 0-LAYER | 4039 ($\pm$3.4) | 4010 ($\pm$3.3) | 29.3 ($\pm$0.46) | 8.73 ($\pm$0.41) | 0.541 ($\pm$0.091) | 0.931 ($\pm$0.043) |
| 1-LAYER | 4022 ($\pm$3.7) | 4003 ($\pm$3.7) | 19.3 ($\pm$0.35) | 6.30 ($\pm$0.49) | 0.538 ($\pm$0.105) | 0.946 ($\pm$0.043) |
| 2-LAYER | 4018 ($\pm$3.3) | 3999 ($\pm$3.3) | 18.5 ($\pm$0.30) | 5.94 ($\pm$0.41) | 0.574 ($\pm$0.096) | 0.978 ($\pm$0.027) |
| 3-LAYER | 4020 ($\pm$3.0) | 4002 ($\pm$2.9) | 18.3 ($\pm$0.38) | 5.73 ($\pm$0.31) | 0.659 ($\pm$0.123) | 0.979 ($\pm$0.037) |

**Table 9: Effect of a pre-broadcast MLP on the Spatial Broadcast VAE's performance, 3D Object-in-Room dataset.** This table is analogous to Table 7, except using the 3D Object-in-Room dataset. Here the model seems to over-representation the dataset generative factors (or which there are 6 for this dataset) without a pre-broadcast MLP (top row). However, adding a pre-broadcast MLP with 2 or 3 layers gives rise to accurate reconstructions with the appropriate number of used latents and good disentangling. Adding a pre-broadcast MLP like this is an alternative to increasing the ConvNet depth in the model (shown in Figure 10).

## G.3 DECODER UPSCALE FACTOR

We acknowledge that there is a continuum of models between the Spatial Broadcast decoder and the Deconv decoder. One could interpolate from one to the other by incrementally replacing the convolutional layers in the Spatial Broadcast decoder's network by deconvolutional layers with stride 2 (and simultaneously decreasing the height and width of the tiling operation). Table 10 shows a few steps of such a progression, where (starting from the bottom) 1, 2, and all 3 of the convolutional layers in the Spatial Broadcast decoder are replaced by a deconvolutional layer with stride 2. We see that this hurts disentangling without affecting the other metrics, further evidence that upscaling deconvolutional layers are bad for representation learning.

| MLP | ELBO | NLL | KL | LATENTS USED | MIG | FACTOR VAE |
|---|---|---|---|---|---|---|
| 0 UPSCALES | 329 ($\pm$4.1) | 305 ($\pm$4.6) | 24.4 ($\pm$0.54) | 7.22 ($\pm$0.34) | 0.147 ($\pm$0.057) | 0.208 ($\pm$0.084) |
| 1 UPSCALE | 327 ($\pm$4.4) | 302 ($\pm$4.9) | 24.4 ($\pm$0.55) | 7.29 ($\pm$0.26) | 0.149 ($\pm$0.048) | 0.194 ($\pm$0.026) |
| 2 UPSCALES | 329 ($\pm$4.3) | 304 ($\pm$4.8) | 24.2 ($\pm$0.60) | 7.14 ($\pm$0.42) | 0.122 ($\pm$0.045) | 0.235 ($\pm$0.070) |
| 3 UPSCALES | 330 ($\pm$2.4) | 305 ($\pm$2.7) | 24.2 ($\pm$0.24) | 7.39 ($\pm$0.08) | 0.110 ($\pm$0.032) | 0.182 ($\pm$0.028) |

**Table 10: Effect of upscale deconvolution on the Spatial Broadcast VAE's performance.** These results use the colored sprites dataset. The columns show the effect of repeatedly replacing the convolutional, stride-1 layers in the decoder by deconvolutional, stride-2 layers (starting at the bottom-most layer). This incrementally reduces performance without affecting the other statistics much, testament to the negative impact of upscaling deconvolutional layer on VAE representations.

## H LATENT SPACE GEOMETRY ANALYSIS FOR CIRCLE DATASETS

We showed visualization of latent space geometry on the circles datasets in Figure 4. However, we also conducted the same style experiments on many more datasets and on FactorVAE models. In this section we will present these additional results.

We consider three generative factor pairs: (X-Position, Y-Position), (X-Position, Hue), and (Redness, Greenness). For each such pair we generate a dataset with independently sampled generative factors and two datasets with dependent generative factors (one with a hole in the center and another with a hole in the corner of generative factor space). For each dataset we run VAE and FactorVAE models with both DeConv and Spatial Broadcast decoder. Broadly, our results in the following figures show that the Spatial Broadcast decoder nearly always helps disentangling. It helps most dramatically on the most positional variation (X-Position, Y-Position) and least significantly when there is no positional variation (Redness, Greenness).

Note, however, that even with no position variation, the Spatial Broadcast decoder does seem to improve latent space geometry in the generalization experiments (Figures 18 and 19). We believe this may be due in part to the fact that the Spatial Broadcast decoder is shallower than the DeConv decoder.

Finally, we explore one completely different dataset with dependent factors: A dataset where half the images have no object (are entirely black). This we do to simulate conditions like that in a multi-entity

VAE such as (Nash et al., 2017) when the dataset has a variable number of entities. These conditions pose a challenge for disentangling, because the VAE objective will wish to allocate a large (low-KL) region of latent space to representing a blank image when there is a large proportion of blank images in the dataset. However, we do see a stark improvement by using the Spatial Broadcast decoder in this case.

The reader will note that in many of the figures in this section the MIG does not capture the intuitive notion of disentangling very well. We believe this is because:

- It depends on a choice of basis for the ground truth factors, and heavily penalizes rotation of the representation with respect to this basis. Yet it is often unclear what the correct basis for the ground truth factors is (e.g. RGB vs HSV vs HSL). For example, see the bottom row of Figure 4.
- It is invariant to a folding of the representation space, as long as the folds align with the axes of variation. See the middle row of Figure 4 for an example of a double-fold in the latent space which isn't penalized by the MIG metric.

More broadly, while a number of metrics have been proposed to quantify disentangling, many of them have serious shortcomings and there is as yet no consensus in the literature which to use (Locatello et al., 2018). We believe it is impossible to quantify how good a representation is with a single scalar, because there is a fundamental trade-off between how much information a representation contains and how well-structured the representation is. This has been noted by others in the disentangling literature (Ridgeway and Mozer, 2018; Eastwood and Williams, 2018; Suter et al., 2018). This disentangling-distortion trade-off is a recapitulation of the rate-distortion trade-off (Alemi et al., 2017) and can be seen first-hand in Figure 3. We would like representations that both reconstruct well and disentangle well, but exactly how to balance these two factors is a matter of subjective preference (and surely depends on the dataset). Any scalar disentangling metric will implicitly favor some arbitrary disentangling-reconstruction potential.

Due to the subjective nature of disentangling and the difficulty in defining appropriate metrics, we put heavy emphasis on latent space visualization as a means for representational analysis. Latent space traversals have been extensively used in the literature and can be quite revealing (Higgins et al., 2017a;b). However, in our experience, traversals suffer two shortcomings:

- Some latent space entanglement can be difficult for the eye to perceive in traversals. For example, a slight change in brightness in a latent traversal that represents changing position can easily go unnoticed.
- Traversals only represent the latent space geometry around one point in space, and cross-referencing corresponding traversals between multiple points is quite time-consuming.

Consequently, we caution the reader against relying too heavily on traversals when evaluating latent space geometry. In many cases, we found the latent factor visualizations in this section to be much more informative of representational quality.

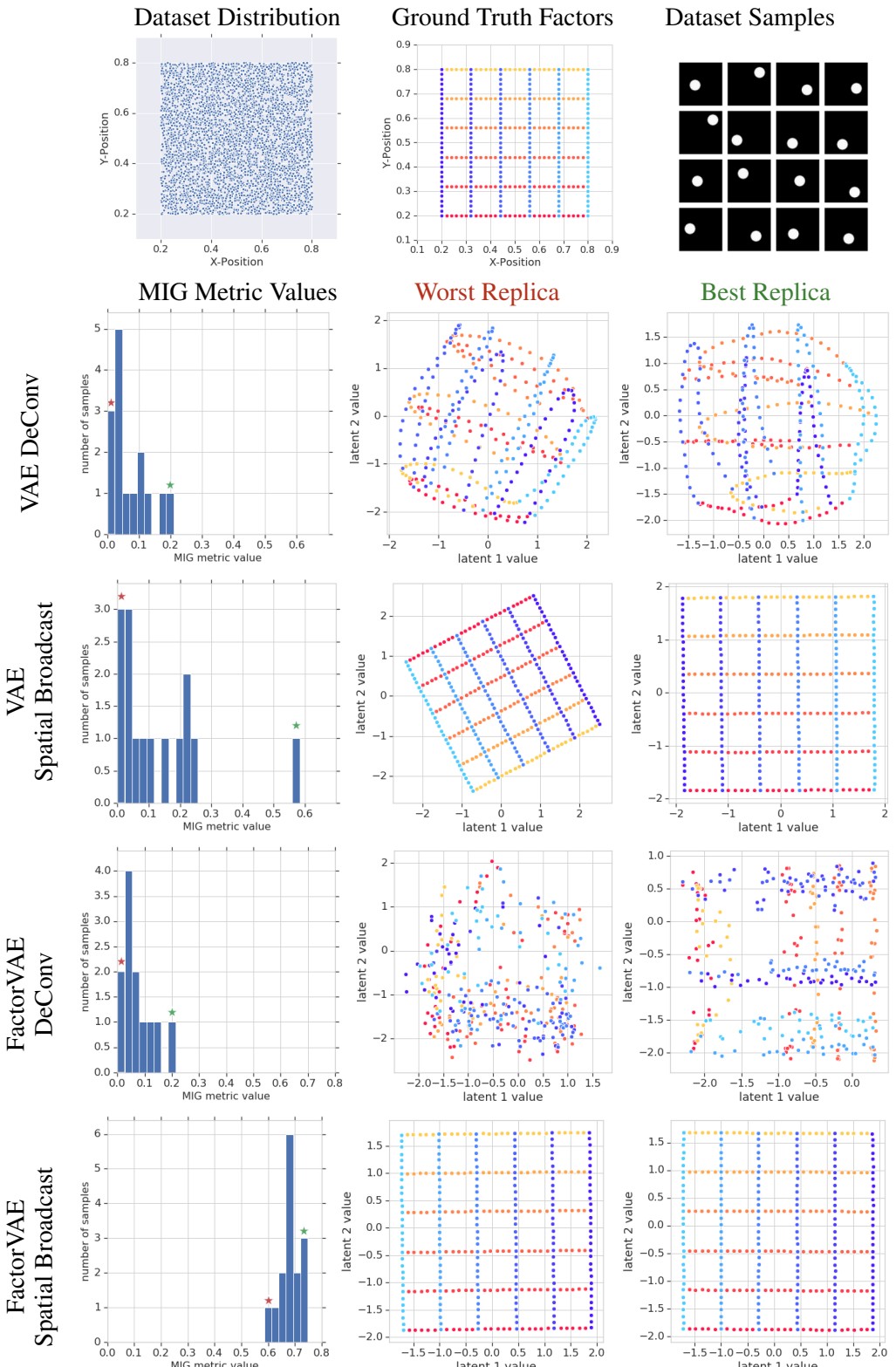

**Figure 11: Latent space analysis for independent X-Y dataset.** This is the same as in Figure 4, except with the additional FactorVAE results. We see that the Spatial Broadcast decoder improves FactorVAE as well as a VAE (and in the FactorVAE case seems to always be axis-aligned). Note that DeConv FactorVAE has a surprisingly messy latent space — we found that using a fixed-variance Normal (instead of Bernoulli) decoder distribution improved this significantly, though still not to the level of the Spatial Broadcast FactorVAE. We also noticed in small-scale experiments that including shape or size variability in the dataset helped FactorVAE disentangle as well. However, FactorVAE does seem to be generally quite sensitive to hyperparameters (Locatello et al., 2018), as adversarial models often are.

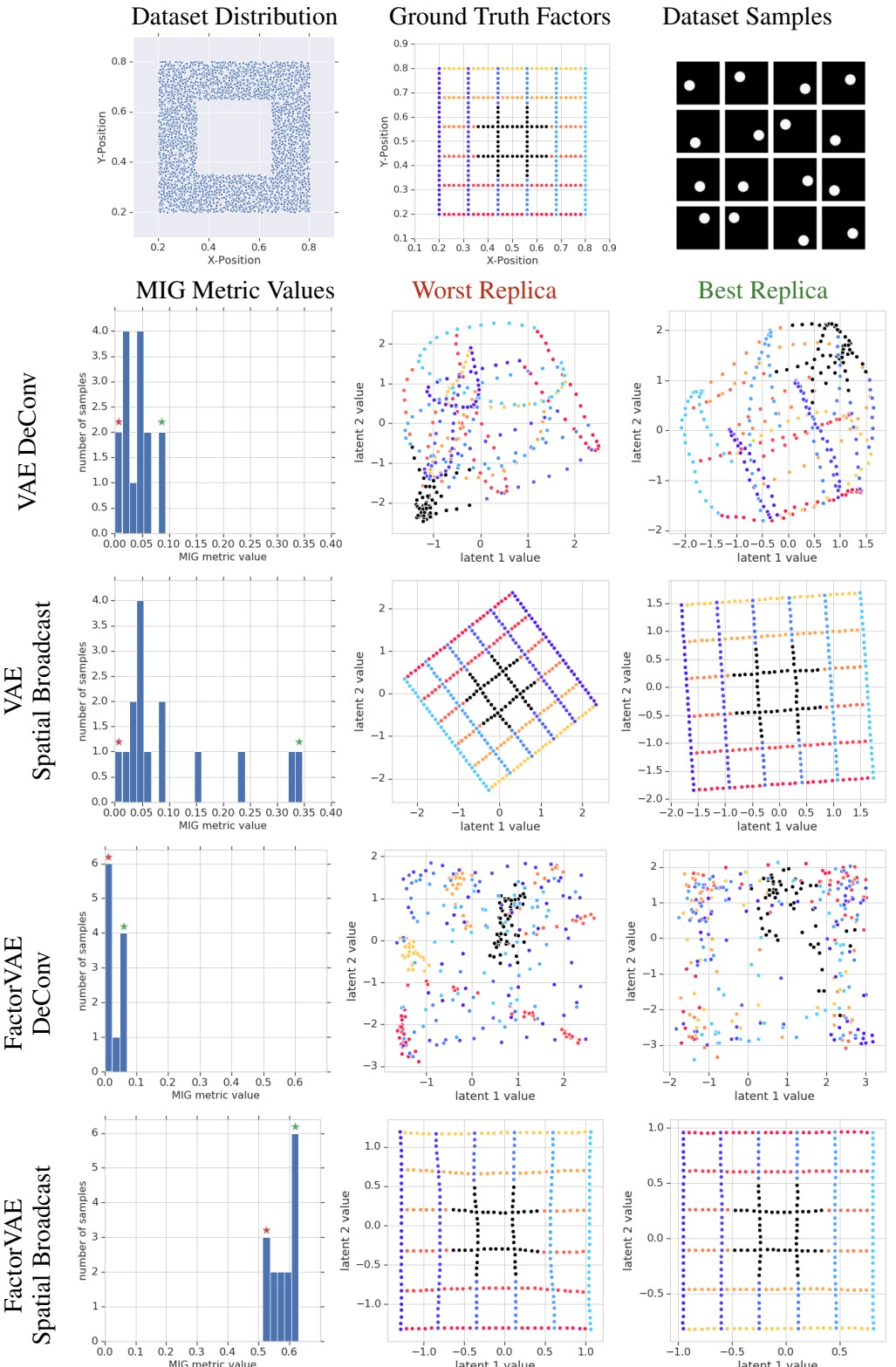

**Figure 12: Latent space analysis for dependent X-Y dataset.** Analogous to Figure 11, except the dataset has a large held-out hole in generative factor space (see dataset distribution in top-left), hence the generative factors are not independent. For the latent geometry visualizations, we do evaluate the representation of images in this held-out hole, which are shown as black dots. This tests for generalization, namely extrapolation in pixel space and interpolation in generative factor space. Again, the Spatial Broadcast decoder dramatically improves the representation — its representation looks nearly linear with respect to the ground truth factors, even through the extrapolation region.

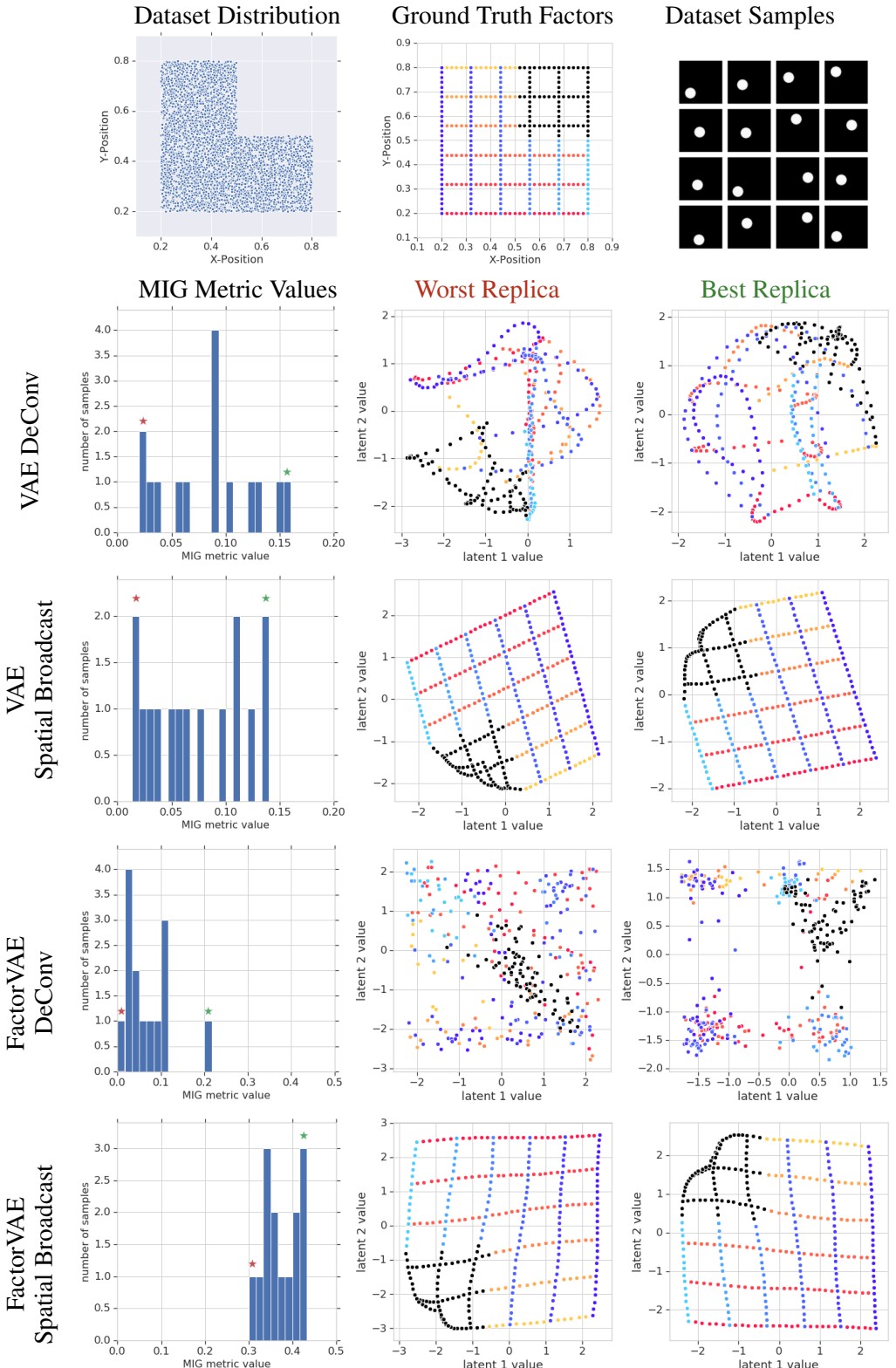

**Figure 13: Latent space analysis for dependent X-Y dataset.** This is similar to Figure 12, except the "hole" in the dataset is in the corner of generative factor space rather than the middle. Hence this tests not only extrapolation in pixel space, but also extrapolation in generative factor space. As usual, the Spatial Broadcast decoder helps a lot, though in this case the extrapolation is naturally more difficult than in Figure 12.

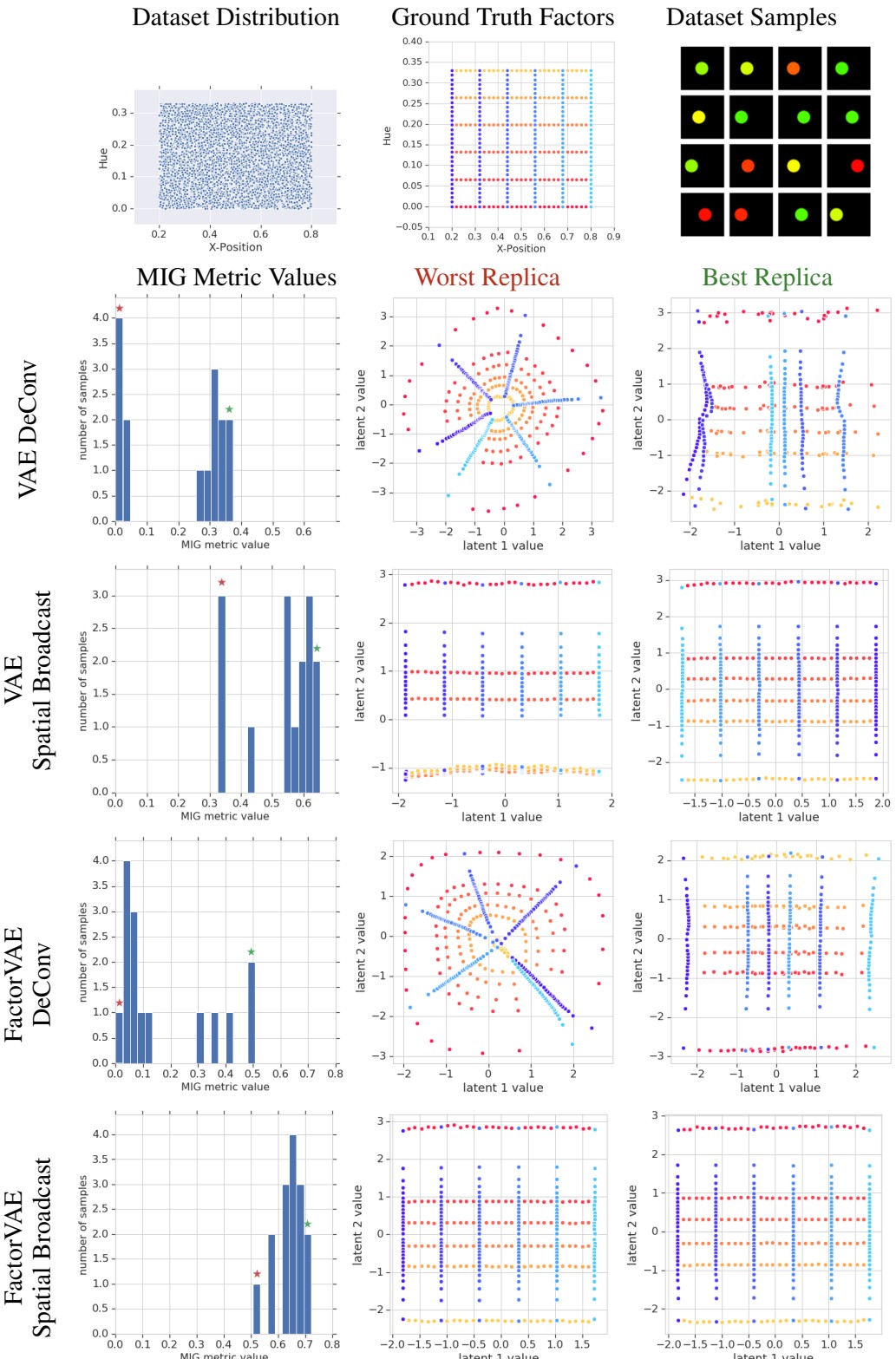

**Figure 14: Latent space analysis for independent X-H dataset.** In this dataset the circle varies in X-Position and Hue. As expected given this has less positional variation than the X-Y datasets, we see the relative improvement of the Spatial Broadcast decoder to be lower, though still quite significant. Interestingly, the representation with the Spatial Broadcast decoder is always axis-aligned and nearly linear in the positional direction, though non-linear in the hue direction. While this is not what the VAE objective is pressuring the model to do (the VAE objective would like to balance mean and variance in its latent space), we attribute this to the fact that a linear representation is much easier to compute from the coordinate channels with ReLU layers than a non-linear effect, particularly with only three convolutional ReLU layers. In a sense, the inductive bias of the architecture is overriding the inductive bias of the objective function.

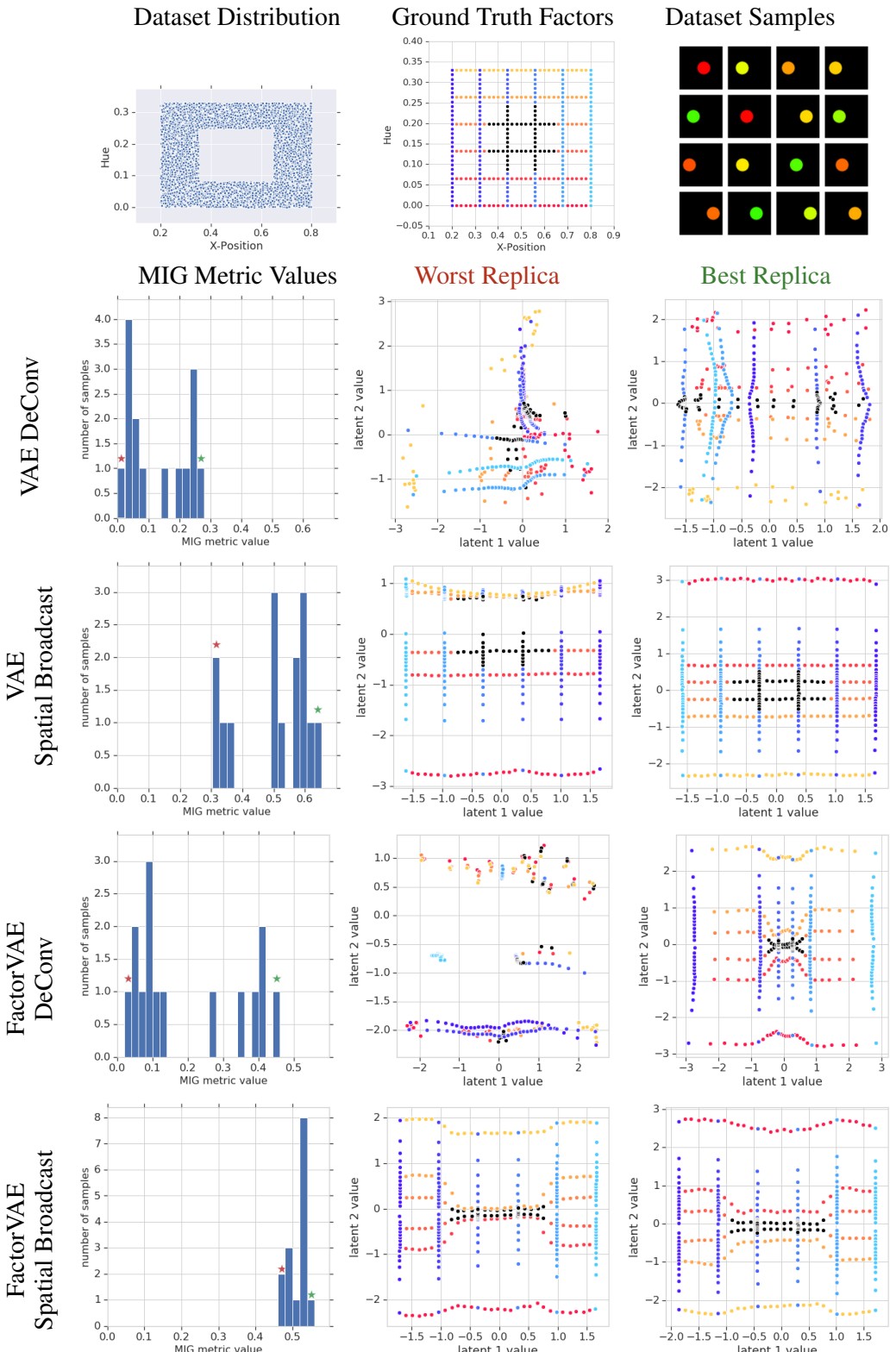

**Figure 15: Latent space analysis for dependent X-H dataset.** This is the same as Figure 14 except the dataset has a held-out "hole" in the middle, hence tests the model's generalization ability. This generalization is extrapolation in pixel space yet interpolation in generative factor space. This poses a serious challenge for the DeConv decoder, and again the Spatial Broadcast decoder helps a lot. Interestingly, note the severe contraction by FactorVAE of the "hole" in latent space. The independence pressure in FactorVAE strongly tries to eliminate unused regions of latent space.

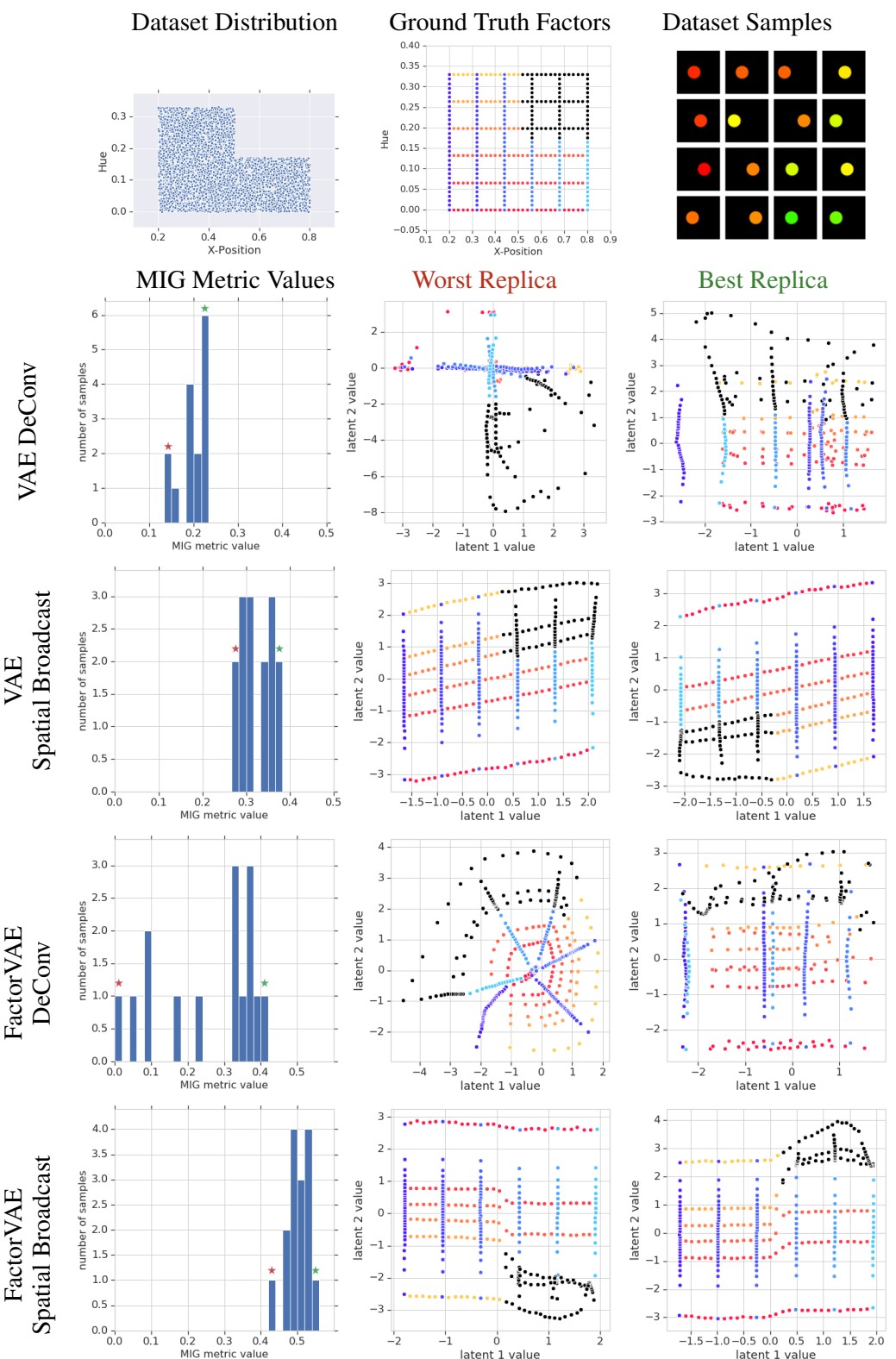

**Figure 16: Latent space analysis for dependent X-H dataset.** This is the same as Figure 15 except the held-out "hole" is in the corner of generative factor space, testing extrapolation in both pixel space and generative factor space. The Spatial Broadcast decoder again yields significant improvements, and as in Figure 15 we see FactorVAE clearly sacrificing latent space geometry to remove the "hole" from the latent space prior distribution (see bottom row).

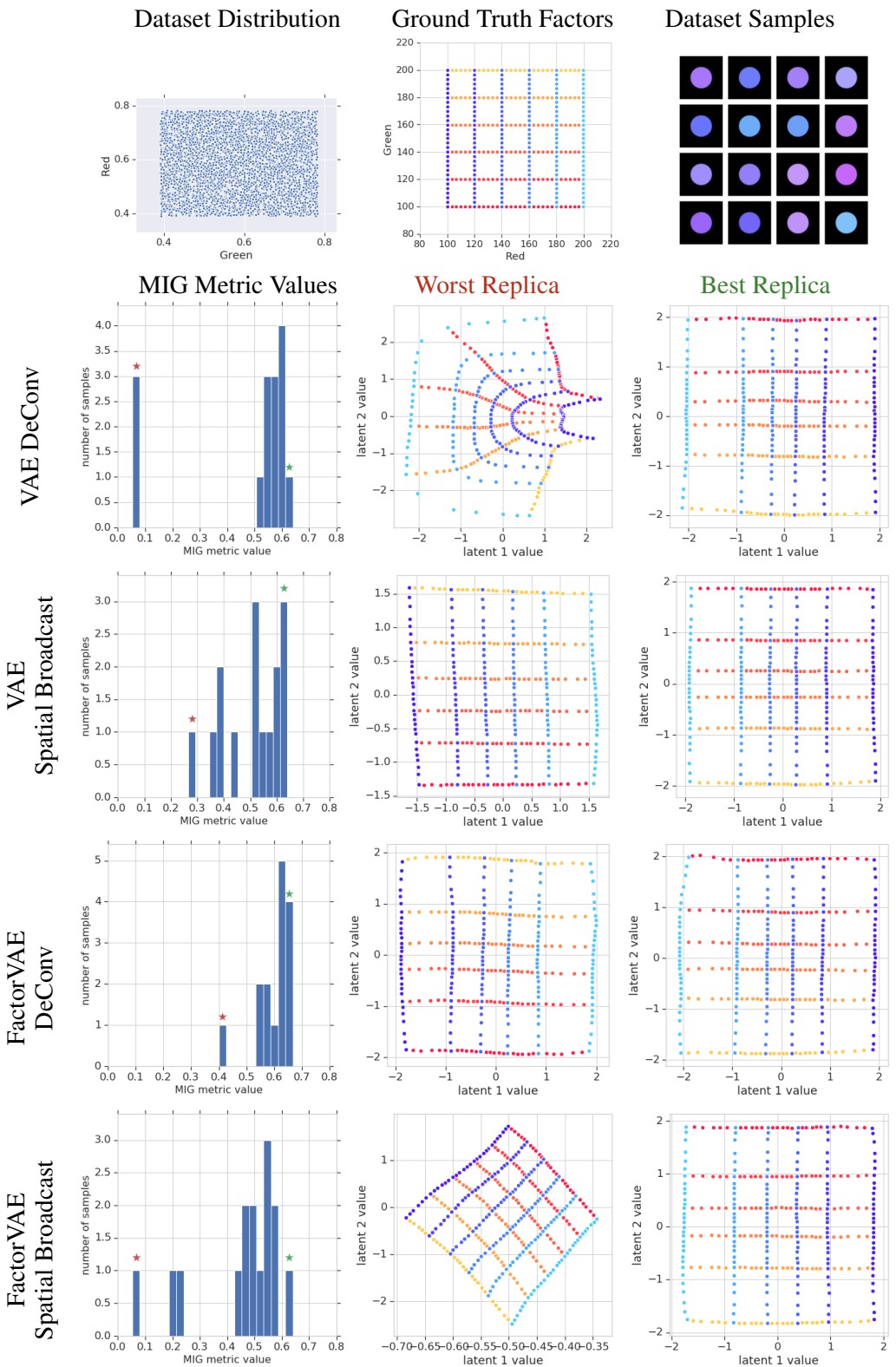

**Figure 17: Latent space analysis for independent R-G dataset.** In this dataset the circles vary only in their Redness and Greenness, not in their position. Here the benefit of the Broadcast decoder is less clear. It doesn't seem to hurt, and may help avoid a small fraction of poor seeds (3 out of 15 with MIG near zero) in a standard VAE.

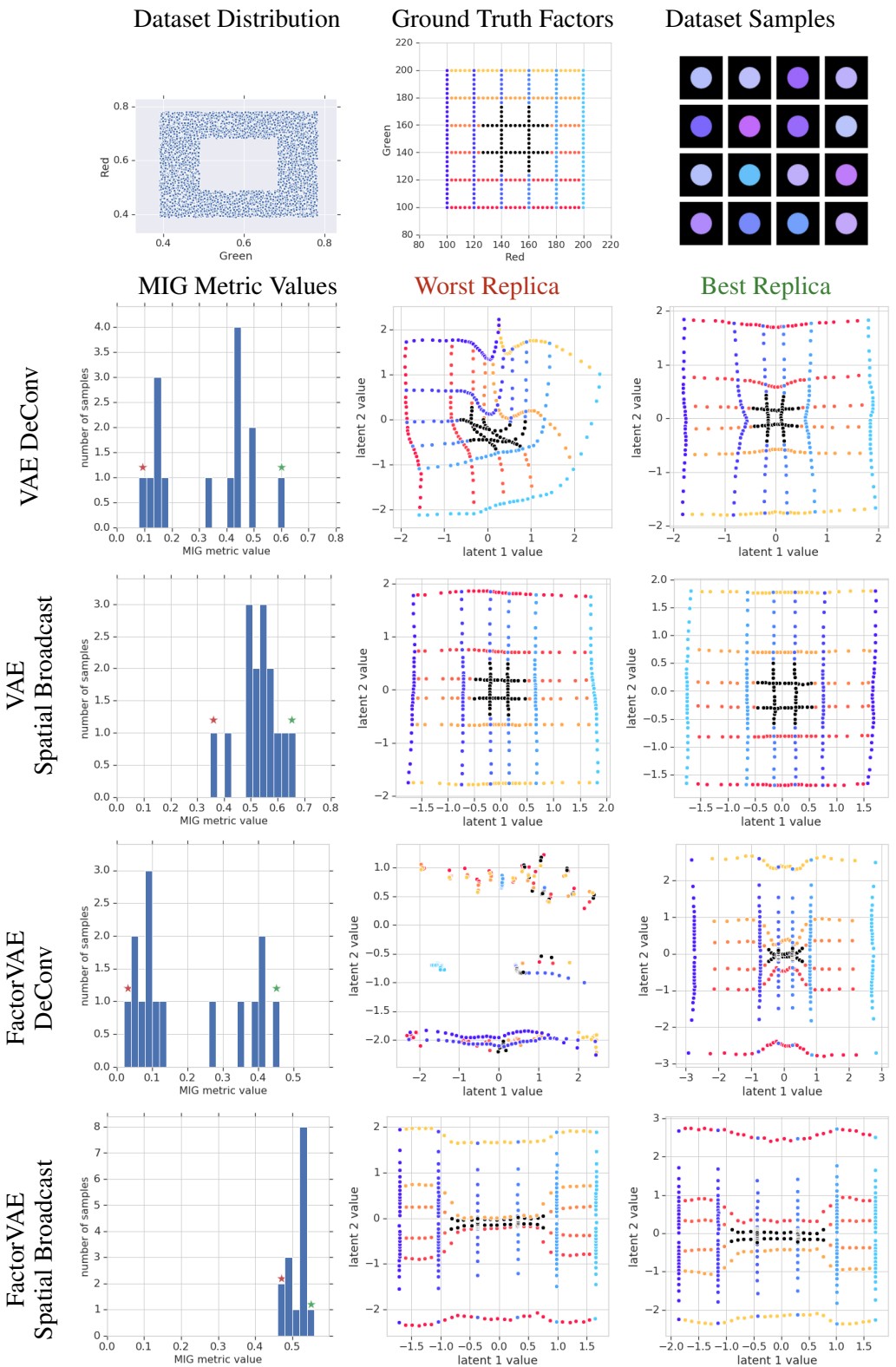

**Figure 18: Latent space analysis for dependent R-G dataset.** This is the same as Figure 17 except with a held-out "hole" in the center of generative factor space, testing extrapolation in pixel space (interpolation in generative factor space). Here the benefit of the Spatial Broadcast decoder is more clear than in Figure 17. We attribute it's benefit here in part to it being a shallower network.

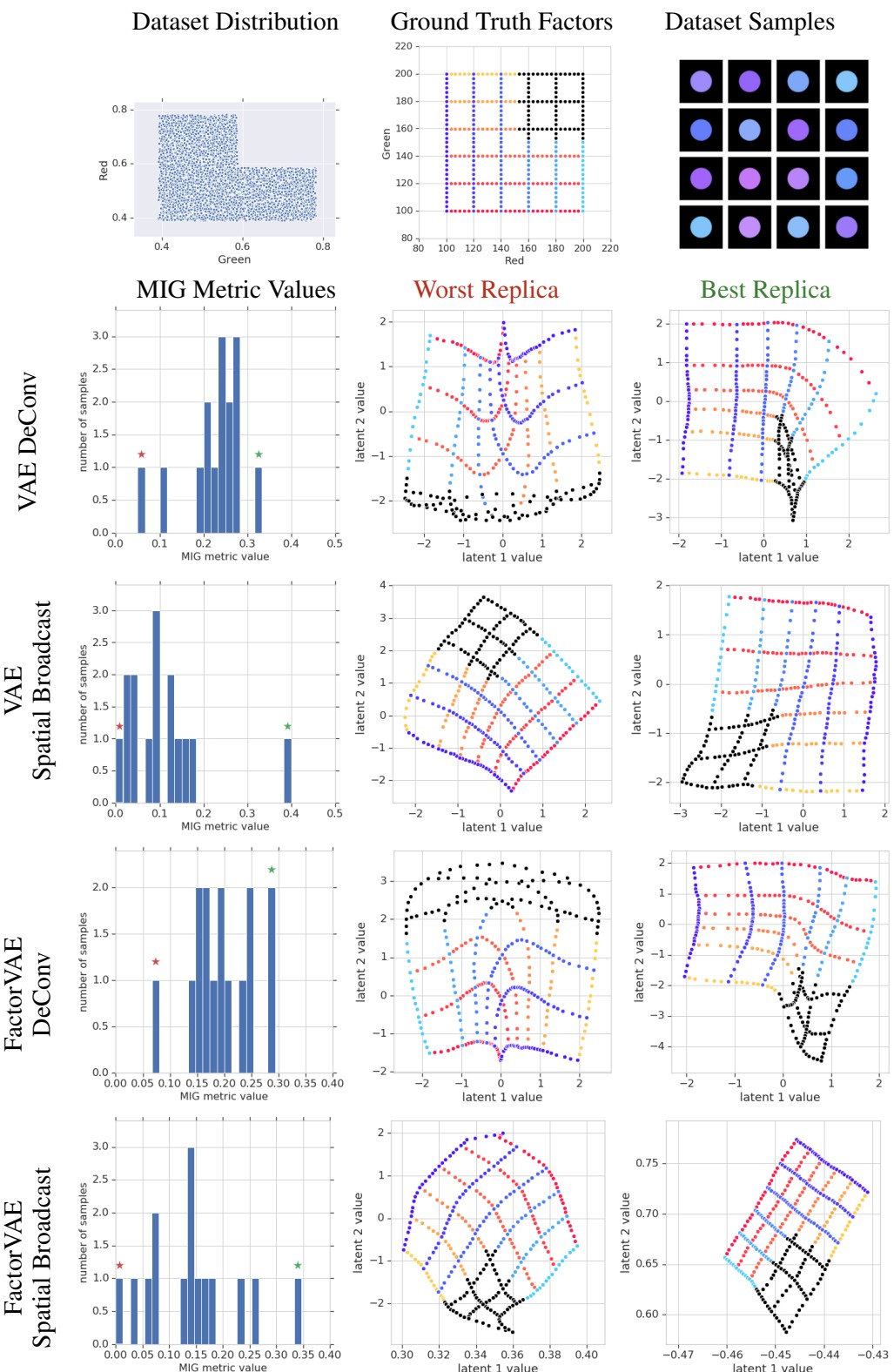

**Figure 19: Latent space analysis for dependent R-G dataset.** This is the same as Figure 18 except the "hole" is in the corner of generative factor space, testing extrapolation in both pixel space and generative factor space. While the Spatial Broadcast decoder seems to give rise to slightly lower MIG scores, this appears to be from rotation in the latent space. It looks like if anything the Spatial Broadcast decoder reduces warping of the latent space geometry, which is what we really care about in the representations.

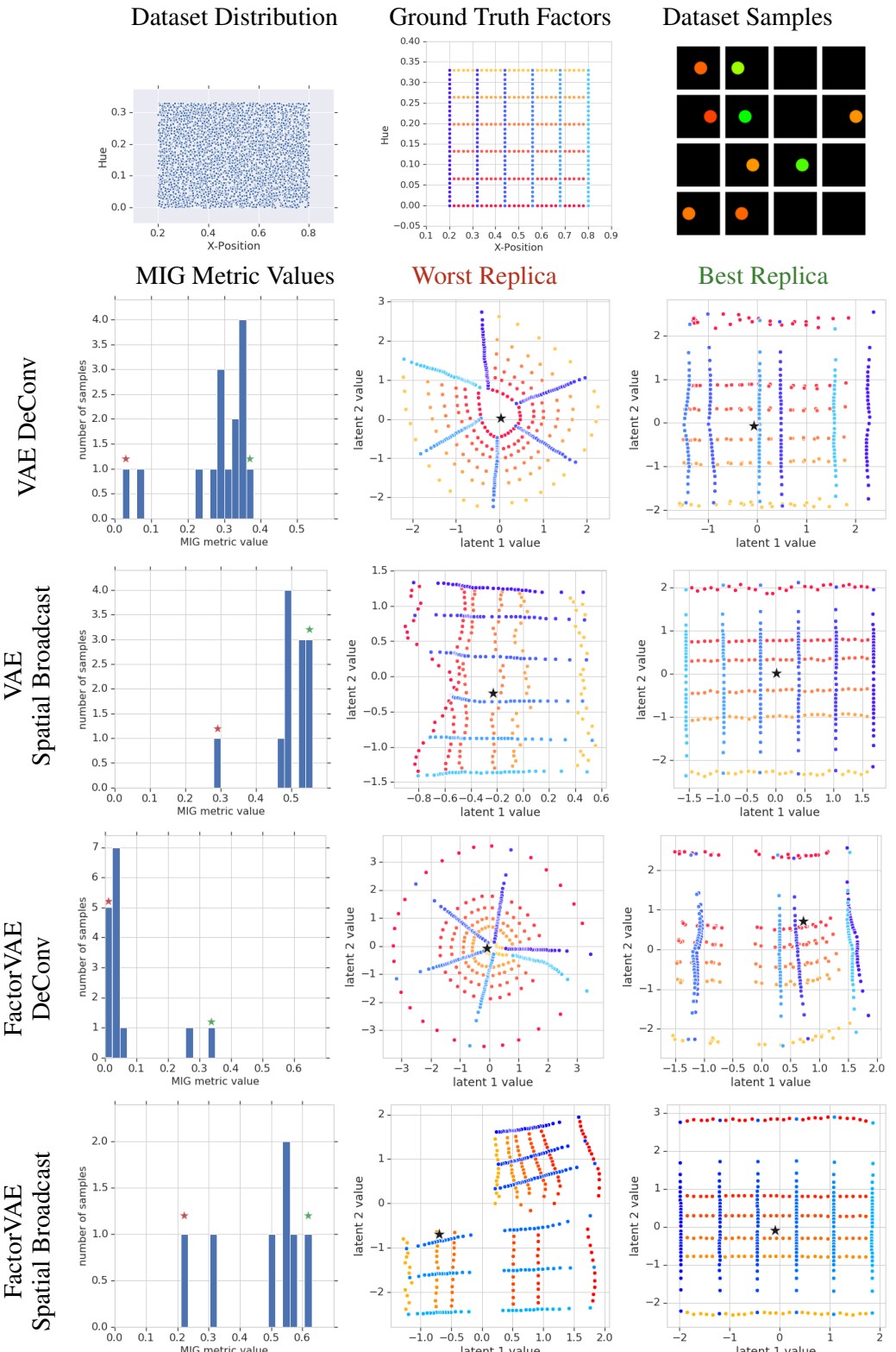

**Figure 20: Latent space analysis for X-H dataset with blank images.** This is the same as Figure 14 except the dataset consists half of images with no objects (entirely black images, as can be seen in the dataset samples in the top-right). This simulates the data distribution produced by the VAE of a multi-entity VAE (Nash et al., 2017) on a dataset with a variable number of objects. Again the Spatial Broadcast decoder improves latent space geometry, according to both the MIG metric and the traversals. In the latent geometry plots, the black star represents the encoding of a blank image. We noticed that the Spatial Broadcast VAE always allocates a third relevant latent to indicate the (binary) presence or absence of an object, a very natural representation of this dataset.

