# OpenReview forum: "Spatial Broadcast Decoder: A Simple Architecture for Disentangled Representations in VAEs"
_ICLR.cc/2019/Workshop/LLD — LLD 2019_

### Official Review · AnonReviewer1 · 2019-04-07
**An Interesting Analysis of Coordinate Tiled VAE Decoders**

**Rating:** 3
**Confidence:** 3

**Review:**

Summary: the authors present a simple extension of VAEs (and CoordConv VAEs) and demonstrate through a variety of experiments that the proposed tiling and (1x1) coord-conv solution produces a more disentangled representation. The presentation of detailed ablation studies is helpful in understanding exactly what benefits are brought by 1x1 convolutions vs. upsampling The empirical results are strong and promising, but a few points should be addressed in the final version.

Major:
  - The results comparing Spatial Broadcast VAEs to CoordConv VAEs is a pretty critical result and should be moved into the main text from the appendix. Note that this should be present for all experiments, including the ones demonstrating the rate-distortion curves. In addition it would be interesting to contrast the CoordConv VAE with a few upsample layers, followed by 1x1 convolutions (as in the Spatial Broadcast VAE) to see if the effect is mainly from tiling or from the lack of upsampling blocks.
  - A simple evaluation of disentanglement would be to use a linear classifier on the (mean) posterior sample after the training of the VAE. This would provide a more informative evaluation of (linear) separation in the latent space. This has been done in Associative Compressive Networks by Alex Graves for example.

Minor:
  - Figure labeling (i.e. a, b) missing on figure 3.
  - Consistency between letter figure labeling and left/right.
  - A4: what is the condition for termination of training? Early Stopping? If so what are the hyper-parameters used there?

---

### Official Review · AnonReviewer2 · 2019-04-07
**Good work and write-up but unlcear about fit with this workshop**

**Rating:** 4
**Confidence:** 1

**Review:**

Pros:
- extensive and thorough experimentation
- interesting and original idea
- proposed an approach that is complimentary to previous approaches and helps improve SOTA results
- comprehensive supplementary

Cons:
- not immediately clear how this work relates to the limited labels setting

---

### Decision · Program_Chairs · 2019-04-08
**Acceptance Decision**

Accept